# Segment Anything with Precise Interaction

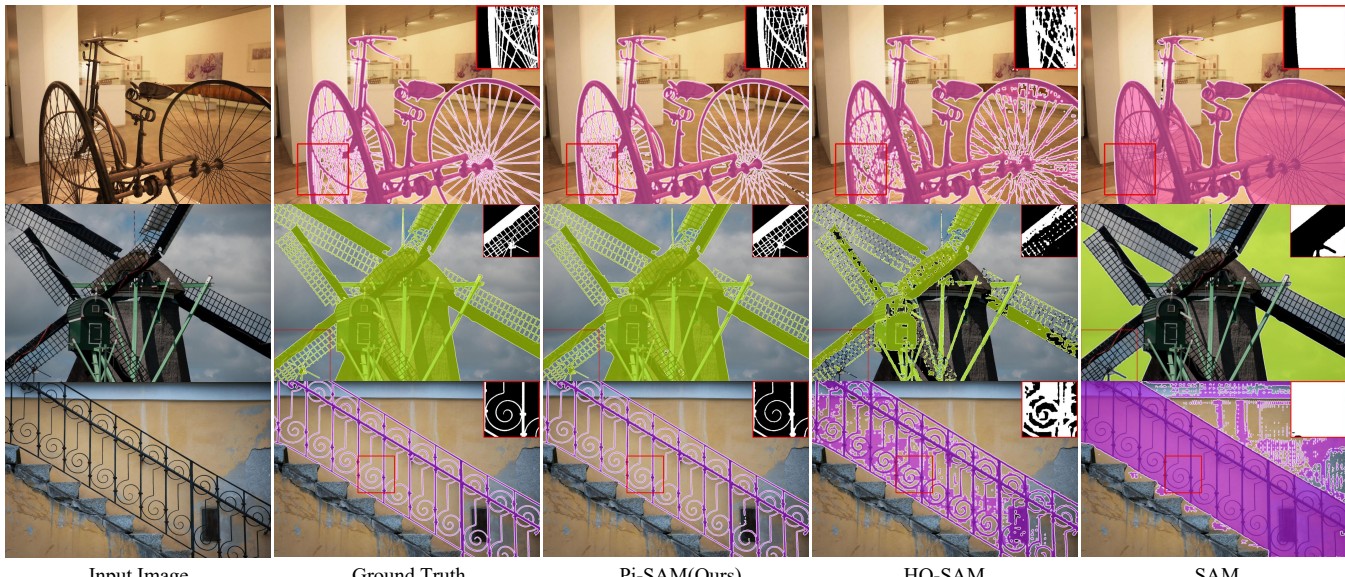

**Figure 1: Qualitative comparisons between the proposed Pi-SAM with SAM and HQ-SAM. In these challenging samples of high-resolution images, our Pi-SAM exhibits a remarkable capability to capture the extremely fine details and perceive the complex topological structures, achieving high-precision segmentation results.**

## ABSTRACT

Although the Segment Anything Model (SAM) has achieved impressive results in many segmentation tasks and benchmarks, its performance noticeably deteriorates when applied to high-resolution images for high-precision segmentation, limiting it's usage in many real-world applications. In this work, we explored transferring SAM into the domain of high-resolution images and proposed Pi-SAM. Compared to the original SAM and its variants, Pi-SAM demonstrates the following superiorities: **Firstly**, Pi-SAM possesses a strong perception capability for the extremely fine details in high-resolution images, enabling it to generate high-precision segmentation masks. As a result,Pi-SAM significantly surpasses previous methods in four high-resolution datasets. **Secondly**, Pi-SAM supports more precise user interactions. In addition to the native promptable ability of SAM, Pi-SAM allows users to interactively refine the segmentation predictions simply by clicking. While the original SAM fails to achieve this on high-resolution

Permission to make digital or hard copies of all or part of this work for personal or classroom use is granted without fee provided that copies are not made or distributed for profit or commercial advantage and that copies bear this notice and the full citation on the first page. Copyrights for components of this work owned by others than the author(s) must be honored. Abstracting with credit is permitted. To copy otherwise, or republish, to post on servers or to redistribute to lists, requires prior specific permission and/or a fee. Request permissions from permissions@acm.org.

*ACM MM, 2024, Melbourne, Australia*

© 2024 Copyright held by the owner/author(s). Publication rights licensed to ACM.
ACM ISBN 978-x-xxxx-xxxx-x/YY/MM
https://doi.org/10.1145/nnnnnnn.nnnnnnn

images. **Thirdly**, building upon SAM, Pi-SAM freezes all its original parameters and introduces very few additional parameters and computational costs to achieve the above performance. This ensures highly efficient model fine-tuning while also retaining the powerful semantic information contained in the original SAM.

## CCS CONCEPTS

• **Computing methodologies → Image segmentation**.

## KEYWORDS

Segment Anything, High-Resolution Segmentation, Dichotomous Image Segmentation, Interactive Segmentation

## 1 INTRODUCTION

High-precision segmentation[13, 22, 29, 33] plays an important role in many vision-centric multimedia systems, such as robotic perception[10, 35], augmented reality [11],and image/video manipulation [7, 15], among others. Compared to the extensively researched visual segmentation tasks such as semantic [19, 26, 30, 38, 41] and instance [1, 16, 18, 31] segmentation, these applications demand higher accuracy on the segmented object boundaries and detailed structures (as shown in Fig. 1), posing greater challenges for the segmentation models. Furthermore, achieving high-precision segmentation often requires making predictions at high resolutions (*2K* or higher), while the cost of annotating this kind of data is prohibitively expensive. Therefore, although several related

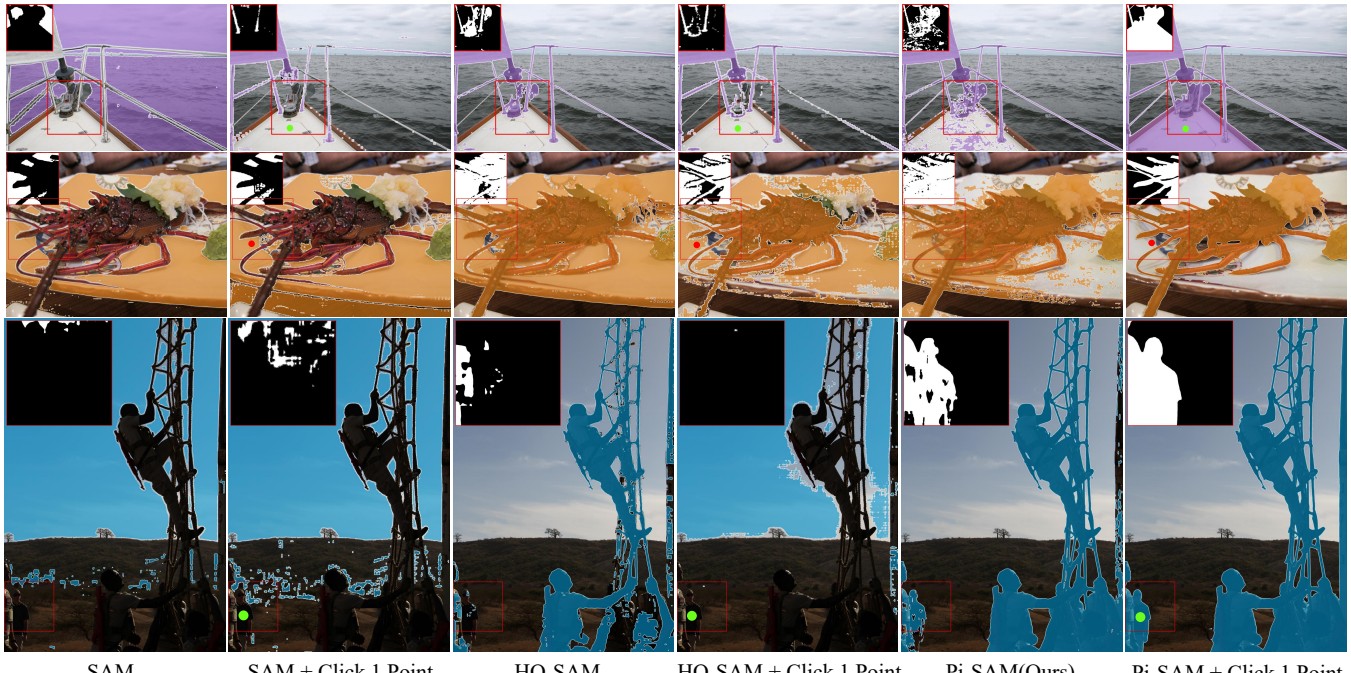

| SAM | SAM + Click 1 Point | HQ-SAM | HQ-SAM + Click 1 Point | Pi-SAM(Ours) | Pi-SAM + Click 1 Point |

**Figure 2: When the model's predictions are imperfect, users may want to guide the model to correct wrongly predicted areas through clicks. Here, green points represent foreground clicks, while red points represent background clicks. It can be observed that both SAM and HQ-SAM fail to correct prediction errors through clicks, whereas our Pi-SAM effectively achieves this.**

datasets [17, 24, 34, 39] have been proposed in previous research, their dataset scales are significantly smaller compared to those of low-resolution segmentation data. This results in the methods tailored for these datasets potentially overfitting to the biases of the datasets, limiting their capabilities in real-world applications.

Recently, the Segment Anything Model (SAM) [14], which is proposed as a foundational model for image segmentation, provides a potential solution to this problem. Due to its extensive training data (with billion-scale masks), SAM exhibits powerful zero-shot capabilities across multiple benchmarks and in-the-wild images. It is an intuitive idea to transfer such large-scale foundational model to the domain of high-resolution images, in order to address the aforementioned issue of insufficient data.

However, in our observation, SAM encounters the following three problems when applied to high-resolution images. **Firstly**, SAM struggles to segment challenging structures. SAM tends to segment objects into large, contiguous regions. When the target object exhibits more complex topological structures, SAM struggles to differentiate between the target object and the background, especially in objects that contain thin structures. **Secondly**, SAM fails to correct incorrect predictions in high-resolution images through multiple interactions. The prompt-driven manner makes SAM allow users to correct the errors of the previous-round predictions by adding new clicks, which has proven effective on low-resolution images. However, on high-resolution images, more interactions often do not improve the results. Instead, as shown in Fig. 2, they lead to worse or even collapsed results. **Thirdly**, SAM's predictions

exhibit noticeable jaggedness and offset along the boundaries on high-resolution images.

We summarize the above problems into two main deficiencies of SAM's architecture: **1) Insufficient output resolution**: The original prediction size of SAM is only $256 \times 256$, which is too small for images with resolutions of *2K* or higher, making it difficult to predict thin structures and accurate boundaries. **2) Interaction on too small size**: In SAM, the information interaction between the image and the prompts is achieved through cross-attention on a $64 \times 64$-sized feature map. On such a small-scale feature map, the detailed structures of the foreground and the surrounding background are mixed together and represented by a single pixel. This results in the model struggling to distinguish which part the user's click refers to, making it incapable to effectively correct the detailed predictions.

To address these two deficiencies, we propose Pi-SAM, which effectively expands SAM's ability to predict and interact at high resolutions. An overview of Pi-SAM's framework is illustrate in Fig. 3 Building upon SAM, our Pi-SAM keeps all the modules of SAM frozen to avoid knowledge forgetting and achieves efficient fine-tuning, and proposes the following two additional modules: **1) A lightweight High-Resolution Mask Decoder**: This module can increase the sizes of the predictions from $256 \times 256$ to $1024 \times 1024$ with low computational cost. Based on the original mask decoder,the HR Mask Decoder merges both the semantic information of low-resolution predictions and the low-level information of high-resolution images, which can effectively enhance the model's perception ability for fine details. **2) An optional Precise Interactor**: For the model's imperfect predictions, this module

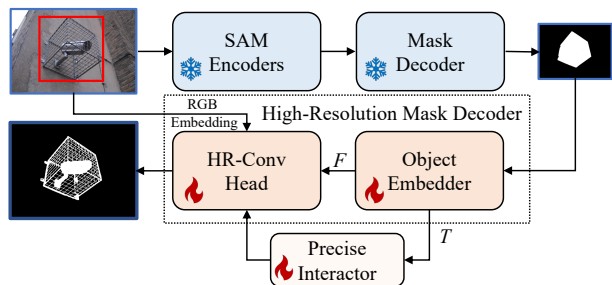

**Figure 3: An overview of the proposed Pi-SAM. We freeze all the original parameters of SAM and propose two additional modules: a lightweight High-Resolution Mask Decoder and an optional Precise Interactor. The former can effectively increase the prediction resolution of SAM, while the latter allows users to interactively indicate the prediction errors and then correct them.**

allows users to indicate wrongly-predicted areas simply by clicking and then corrects the errors. Specifically, it encodes both the semantic and positional information of the user-clicked points and uses such information to update the image features to obtain corrected predictions.

To evaluate the performance of the proposed Pi-SAM, we further conduct both qualitative and quantitative comparisons between Pi-SAM and previous methods, including SAM, SAM's variant, and the methods specifically tailored for high-resolution images, in four high-resolution datasets. Experimental results demonstrate the superiority of Pi-SAM in the following three main aspects:

**Firstly**, Pi-SAM possesses a strong perception capability for the extremely fine details in high-resolution images, enabling it to generate high-precision segmentation masks. As a result, Pi-SAM significantly surpasses previous methods in the four datasets.

**Secondly**, Pi-SAM supports more precise user interactions. In addition to the native promptable ability of SAM, Pi-SAM allows users to interactively refine segmentation predictions through simply clicking. However, the original SAM fails to achieve this on high-resolution images.

**Thirdly**, building upon SAM, Pi-SAM freezes all its original parameters and introduces very few additional parameters and computational costs to achieve the above two points. This ensures highly efficient model transfer while also retaining the powerful semantic information contained in the original SAM.

We believe the above results effectively demonstrate the powerful capability of the proposed Pi-SAM in high-resolution image segmentation, and hope that Pi-SAM can serve as a robust general segmentation tool for high-resolution images and realize its value across various downstream applications.

## 2 RELATED WORK

### 2.1 Segment Anything Models

Segment Anything Model(SAM)[14], as a foundational visual model aiming to perform general image segmentation, has demonstrated remarkable capabilities across various segmentation tasks. Its powerful performance also inspires a series of SAM-based works. Some

of these efforts are dedicated to applying SAM to other downstream tasks, including image editing [9], low-level vision [20, 37], and 3D reconstruction and segmentation [2, 23]. Meanwhile, another part aims to expand the performance of the original SAM, such as transferring SAM to video segmentation and tracking [6, 36], speeding up SAM's inference speed [40, 42] and incorporating semantic information into SAM's category-agnostic manner [21].

Sharing the most similar motivation to ours, HQ-SAM [12] introduces an additional output token and more feature fusion to enhance the segmentation precision of SAM. However, their method still fails to address SAM's deficiencies in interaction size and output resolution,and is thus still unable to achieve satisfactory high-resolution segmentation. Compared to previous works, our Pi-SAM is the first one to focus on improving SAM's output resolution and achieving precise interaction, and achieves excellent results on high-resolution images.

### 2.2 High-Resolution Segmentation

Compared to other segmentation tasks, high-resolution segmentation emphasizes more on the model's ability to segment extremely fine details in high-resolution images, such as mesh objects and thin lines. Previous related methods can be mainly divided into two categories, where one focuses on post-processing existing segmentation results to enhance their accuracy in fine details [4, 25, 29]. While these methods effectively enhance the accuracy of existing segmentation results, the post-processing approach significantly increases computational complexity and inference time consumption. The other type [17, 22, 24, 43] primarily focuses on designing methods tailored for specific high-resolution datasets However, these methods are prone to overfitting the biases in specific datasets, making generalization challenging. On the contrary, we opted to transfer a foundational model to high-resolution images and fine-tune it on multiple high-resolution segmentation datasets jointly. This approach allows for better generalization across different datasets and in-the-wild images.

## 3 METHOD

We propose Pi-SAM, which effectively expands SAM's ability to predict and interact at high resolution. In order to avoid knowledge forgetting and achieve efficient fine-tuning, our Pi-SAM keeps all the modules of SAM frozen, and proposes only two additional modules: **High-Resolution Mask Decoder** and **Precise Interactor**. Following this, we start with a brief review of SAM and its variant, HQ-SAM, in Sec. 3.1. Then, in Sec. 3.2 and Sec. 3.3, we will give detailed descriptions of the two newly proposed modules, respectively. The overall framework of our Pi-SAM is illustrated in Fig. 4.

### 3.1 Preliminaries: SAM and HQ-SAM

The original architecture of SAM consists of a ViT-based [8] image encoder, a prompt encoder and a mask decoder. The image encoder transforms the input $1024 \times 1024$-sized image into a $16\times$ downsampled image embedding. The prompt encoder is employed to encode the points, boxes or masks into prompt tokens. The mask decoder employs several learnable embeddings, termed as output tokens, to extract the object representation through a Transformer [28] block, which updates both the output tokens and the image embedding.

**Figure 4: An overview of the proposed Pi-SAM. We propose two additional modules: a High-Resolution Mask Decoder and a Precise Interactor. The High-Resolution Mask Decoder consists of an Object Embedder and a High-Resolution Convolutional Head (referred to as HR-Conv Head in the figure). The Object Embedder enhances the low-resolution mask features output from SAM. The HR-Conv Head replaces the dot-product-based output layer of SAM to yield high-resolution predictions. The Precise Interactor, an optional module, allows users to identify inaccuracies in predictions simply by clicking on the wrongly-predicted areas, which it then automatically corrects.**

The updated image embedding is then upsampled to $256 \times 256$ as the mask feature and segmentation predictions are made through dot-product between the mask feature and the output tokens.

HQ-SAM adds a new HQ-output token to the original output tokens and incorporates additional low-level information, which is obtained from the shallow feature of ViT, into the above mask feature, thus enhancing the model's perception of low-level features. However, this approach fails to address the deficiencies of the original SAM's interaction size and output resolution being too small; therefore, satisfactory high-resolution segmentation is still not achievable.

## 3.2 High-Resolution Mask Decoder

Due to the inevitable occurrence of edge jagging and offsets when resizing the low-resolution mask to the original image resolution, making predictions at a larger resolution is necessary. As the most intuitive idea, we can simply add more upsampling layers for the mask feature to obtain a larger-size mask feature and then make predictions at the new size. However, in our observation, such a simple approach doesn't lead to a significant improvement in accuracy. Instead, the output high-resolution mask and low-resolution mask don't differ much. Therefore, we propose the High-Resolution Mask Decoder, which can not only improve the resolution of the mask feature, but also effectively enhance the mask feature and boost model's perceptual capability for both high-level semantic information and low-level detailed information.

Specifically, the High-Resolution Mask Decoder consists of two main sub-modules: an **Object Embedder** and an **High-Resolution Convolutional Head**. The Object Embedder is employed to enhance the mask feature output from the original mask decoder of SAM. It takes the mask feature and the predicted low-resolution mask as input, with sizes of $32 \times 256 \times 256$ and $1 \times 256 \times 256$, respectively. These two parts are first concatenated along the channel dimension. Subsequently, the Object Embedder employs several convolutional layers (along with normalization and activation) to embed the coarse semantic information in the low-resolution mask into the mask feature, in order to enhance the mask feature's perception of the target objects.

To further correct the prediction errors in the low-resolution mask, through observation, we found that the detailed structures are often heavily repeated in different spatial locations (such as the mesh-like structure shown in Fig. 4), indicating significant spatial correlations. Therefore, designing a module to increase the receptive field of the mask feature, in order to model the aforementioned spatial correlations between one pixel and its surrounding, is crucial for modeling the detailed structures and correcting prediction errors. To achieve this goal, the simplest approach is to use a transformer-based architecture, which can encode spatial correlations in mask features through self-attention. However, performing self-attention in such a $256 \times 256$-size feature would result in an unacceptable computational overhead. Since we aim for fewer additional parameters and efficient fine-tuning, we have abandoned

the transformer-based approaches. Instead, we employ a module similar to the ASPP (Atrous Spatial Pyramid Pooling) [3], which consists of three parallel branches: a $3 \times 3$ Deformable Convolution, a $7 \times 7$ Deformable Convolution, and a skip connection. Such multiple parallel branches can effectively model the correlations between each pixel in the mask feature and its neighboring pixels at different distances.

Following the feature enhancement of Object Embedder, we propose a lightweight High-Resolution Convolutional Head to make high-resolution prediction. It first employs several Transposed Convolution layers to upsample the enhanced mask feature from $256 \times 256$ to $1024 \times 1024$. To further boost the model's ability to capture fine details, we incorporate an additional RGB embedding into this upsampled feature. It is obtained directly from the original image through two convolutional layers, representing the low-level information such as color and texture. Finally, we replace the dot-product-based output layer of original SAM with two convolutional layers. This is because at high resolution, dot product outputs may lead to some local discontinuities, while convolutional outputs can better model the correlation between adjacent pixels.

## 3.3 Precise Interactor

SAM allows users to refine the previous predictions by incorporating additional clicks, which is achieved by using the mask from the previous prediction along with new click points as the prompt for the next round of prediction. However, this pipeline does not work on high-resolution images, and usually leads to worse or even collapsed results. To facilitate SAM with the capabilities of high-precision interaction and detailed error correction, we propose an optional Precise Interactor. It takes as input the mask feature enhanced by the Object Embedder, the output token of the original mask decoder, and the points clicked by users. As the clicked points may include two cases: incorrectly predicting background as foreground and vice versa, this module also employs two learnable embeddings to represent the properties of the input points (shown as "Flag Embedding" in Fig. 4).

The Precise Interactor first samples the feature based on the coordinates of each input point from the mask feature. Then, the feature of each point, along with the corresponding flag embedding and positional embedding, are concatenated along the channel dimension. Afterward, an MLP is employed to fuse the information from each part, resulting in the final representation for each point, termed as point token. In order to encode this information into the mask feature to effectively correct the final prediction, we modified the bidirectional Transformer block in SAM to facilitate the information propagation between the mask feature and the point tokens. The point tokens are first concatenated with the output token along the length dimension, as the input tokens. Subsequently, the bidirectional Transformer block performs both image-to-token and token-to-image attention, to embed both the target object information in the output token and the user-clicking information in the point tokens into the mask feature.

To obtain the input points during model training, we designed a process that simulates user clicking. We first find the wrongly predicted areas by comparing the difference between the current prediction and the ground truth. Then we sample one point from each connected component in the wrongly predicted areas. Detailed pipeline can be found in the supplementary material.

## 4 EXPERIMENTS

In the following, we will first describe the implementation details in Sec. 4.1. Then, we will dive into the quantitative results of different segmentation tasks from Sec. 4.2 to Sec. 4.4. In Sec. 4.5, we will conduct ablation studies to evaluate the model efficiency and the effectiveness of the additional modules we proposed.

## 4.1 Implementation Details

*4.1.1 **Training Scheme**.* Due to the limited scale of high-resolution segmentation datasets, we collected a combined high-resolution dataset to train the proposed Pi-SAM. The combined dataset includes four commonly used datasets from three segmentation tasks: DIS5K [24] from Dichotomous Image Segmentation, HRSOD [39] and UHRSD [34] from Salient Object Detection, and the ThinObject5K [17] from Interactive Segmentation.

Based on the ViT-base, ViT-large, and ViT-huge versions of SAM, we trained three versions of Pi-SAM. All three versions maintained the same configurations and were trained for 100 epochs on the combined dataset mentioned above, with a cosine-decay learning rate schedule. Since we froze all the original parameters of SAM and introduced additional lightweight modules, compared to SAM, our Pi-SAM requires far fewer computational resources to complete training. All experiments of Pi-SAM were conducted using 8 NVIDIA 4090 GPUs. And the global batch size was set to 32. For more training details, please refer to our supplementary material.

Among the three types of prompts supported by SAM, including points, boxes, and masks, a single point struggles to represent complex objects in high-resolution images, while mask input requires more manual effort from the user and is less practical. Therefore, during the training process, we used the boxes as the input prompts to train our Pi-SAM, which were derived from the ground truth segmentation masks.

*4.1.2 **Evaluation Setup**.* We conduct experiments on multiple high-resolution benchmarks to evaluate different aspects of Pi-SAM's performance. In order to evaluate Pi-SAM's generalization ability, experiments on different datasets are all based on the Pi-SAM version trained on the merged dataset mentioned above, without applying any additional training tailored to specific data.

In all experiments, we employ the bounding boxes produced by the ground truth masks as the input prompts for SAM, HQ-SAM and our Pi-SAM. In Sec. 4.2, we conduct experiments on the DIS task to evaluate Pi-SAM's performance on data with highly complex topological structures. Section 4.3 presents experiments on the SOD task to evaluate Pi-SAM's performance on objects that are more salient and have simpler topological structures. In Sec. 4.4, we apply SAM, HQ-SAM, and our Pi-SAM to the BIG dataset [5] to assess the zero-shot capability of the three models on high-resolution images. In all experiments, consistent with SAM and HQ-SAM, we use the ViT-huge version of Pi-SAM as the default model for comparison with previous methods.

Further details about the dataset and experiments are provided in the corresponding sections.

**Table 1: Results on DIS5K dataset. Among the three SAM-based methods, we have bolded the best results of each ViT version. For the comparison of all methods, top 1, 2, and 3 results are highlighted in red, green, and blue, respectively.**

| Dataset | Measure | Baseline PGNet[33] | IS-Net[24] | FP-DIS[44] | Birefnet[43] | UDUN [22] | ViT-b SAM | HQ-SAM | Pi-SAM | ViT-l SAM | HQ-SAM | Pi-SAM | ViT-h SAM | HQ-SAM | Pi-SAM |
|---|---|---|---|---|---|---|---|---|---|---|---|---|---|---|---|
| DIS-TE1 | $F_\beta^x \uparrow$ | .754 | .74 | .784 | .866 | .784 | .72 | .864 | **.89** | .783 | .892 | **.917** | .755 | .895 | **.917** |
| | $F_\beta^\omega \uparrow$ | .68 | .662 | .713 | .829 | .72 | .681 | .839 | **.869** | .746 | .875 | **.902** | .721 | .878 | **.903** |
| | $M \downarrow$ | .067 | .074 | .06 | .036 | .059 | .114 | .034 | **.027** | .09 | .023 | **.02** | .106 | .025 | **.019** |
| | $S_m \uparrow$ | .8 | .787 | .821 | .889 | .817 | .737 | .872 | **.894** | .787 | .897 | **.917** | .766 | .898 | **.92** |
| | $E_\phi^m \uparrow$ | .848 | .82 | .86 | .917 | .864 | .82 | .933 | **.947** | .852 | .952 | **.96** | .833 | .951 | **.961** |
| | $HCE_\gamma \downarrow$ | 162 | 149 | 160 | 116 | 140 | 442 | 196 | **176** | 215 | 184 | **129** | 206 | 192 | **127** |
| DIS-TE2 | $F_\beta^x \uparrow$ | .807 | .799 | .827 | .906 | .829 | .674 | .872 | **.903** | .766 | .892 | **.918** | .708 | .895 | **.924** |
| | $F_\beta^\omega \uparrow$ | .743 | .728 | .767 | .876 | .768 | .627 | .848 | **.887** | .717 | .875 | **.904** | .666 | .877 | **.912** |
| | $M \downarrow$ | .065 | .07 | .059 | .031 | .058 | .149 | .039 | **.027** | .107 | .032 | **.023** | .141 | .032 | **.021** |
| | $S_m \uparrow$ | .833 | .823 | .845 | .913 | .843 | .685 | .875 | **.907** | .756 | .894 | **.918** | .713 | .895 | **.923** |
| | $E_\phi^m \uparrow$ | .88 | .858 | .893 | .943 | .886 | .785 | .939 | **.953** | .831 | .948 | **.959** | .791 | .948 | **.963** |
| | $HCE_\gamma \downarrow$ | 375 | 340 | 373 | 283 | 325 | 809 | 457 | **383** | 465 | 438 | **316** | 460 | 449 | **316** |
| DIS-TE3 | $F_\beta^x \uparrow$ | .843 | .83 | .868 | .92 | .865 | .614 | .862 | **.899** | .687 | .862 | **.912** | .629 | .87 | **.915** |
| | $F_\beta^\omega \uparrow$ | .785 | .758 | .811 | .888 | .809 | .564 | .836 | **.882** | .634 | .84 | **.896** | .583 | .848 | **.901** |
| | $M \downarrow$ | .056 | .064 | .049 | .029 | .05 | .185 | .044 | **.03** | .143 | .042 | **.027** | .176 | .041 | **.024** |
| | $S_m \uparrow$ | .844 | .836 | .871 | .918 | .865 | .634 | .865 | **.901** | .696 | .87 | **.91** | .654 | .873 | **.915** |
| | $E_\phi^m \uparrow$ | .911 | .883 | .922 | .951 | .917 | .735 | .932 | **.953** | .778 | .93 | **.955** | .748 | .933 | **.959** |
| | $HCE_\gamma \downarrow$ | 797 | 687 | 780 | 617 | 658 | 1355 | 907 | **779** | 900 | 882 | **689** | 893 | 894 | **674** |
| DIS-TE4 | $F_\beta^x \uparrow$ | .831 | .827 | .846 | .906 | .846 | .531 | .809 | **.869** | .613 | .802 | **.89** | .576 | .819 | **.891** |
| | $F_\beta^\omega \uparrow$ | .774 | .753 | .788 | .866 | .792 | .497 | .785 | **.855** | .577 | .785 | **.876** | .545 | .799 | **.879** |
| | $M \downarrow$ | .065 | .072 | .061 | .038 | .059 | .251 | .072 | **.046** | .191 | .072 | **.038** | .218 | .066 | **.036** |
| | $S_m \uparrow$ | .841 | .83 | .852 | .902 | .849 | .563 | .817 | **.871** | .639 | .819 | **.885** | .611 | .827 | **.889** |
| | $E_\phi^m \uparrow$ | .899 | .87 | .906 | .94 | .901 | .672 | .899 | **.939** | .734 | .895 | **.949** | .707 | .905 | **.952** |
| | $HCE_\gamma \downarrow$ | 3361 | 2888 | 3347 | 2830 | 2785 | 4045 | 3638 | **3299** | 3482 | 3590 | **3159** | 3488 | 3617 | **3113** |
| DIS-TE(1-4) | $F_\beta^x \uparrow$ | .809 | .799 | .831 | .897 | .823 | .635 | .852 | **.89** | .712 | .862 | **.909** | .667 | .87 | **.912** |
| | $F_\beta^\omega \uparrow$ | .746 | .726 | .77 | .863 | .763 | .592 | .827 | **.873** | .668 | .844 | **.894** | .629 | .85 | **.899** |
| | $M \downarrow$ | .063 | .07 | .047 | .036 | .059 | .175 | .047 | **.033** | .133 | .042 | **.027** | .16 | .041 | **.025** |
| | $S_m \uparrow$ | .83 | .819 | .847 | .905 | .838 | .655 | .857 | **.893** | .72 | .87 | **.907** | .686 | .873 | **.912** |
| | $E_\phi^m \uparrow$ | .885 | .858 | .895 | .937 | .892 | .753 | .926 | **.948** | .799 | .931 | **.956** | .77 | .934 | **.959** |
| | $HCE_\gamma \downarrow$ | 1173 | 1016 | 1165 | 961 | 977 | 1663 | 1300 | **1191** | 1266 | 1274 | **1102** | 1262 | 1288 | **1057** |
| DIS-VD | $F_\beta^x \uparrow$ | .798 | .791 | .823 | .9 | .831 | .654 | .849 | **.883** | .739 | .858 | **.91** | .687 | .865 | **.91** |
| | $F_\beta^\omega \uparrow$ | .733 | .717 | .763 | .865 | .772 | .609 | .825 | **.866** | .698 | .841 | **.897** | .652 | .847 | **.899** |
| | $M \downarrow$ | .067 | .074 | .062 | .034 | .057 | .167 | .046 | **.035** | .117 | .042 | **.026** | .151 | .04 | **.026** |
| | $S_m \uparrow$ | .824 | .813 | .843 | .906 | .844 | .665 | .856 | **.889** | .738 | .868 | **.909** | .7 | .871 | **.912** |
| | $E_\phi^m \uparrow$ | .879 | .856 | .891 | .938 | .892 | .761 | .925 | **.945** | .817 | .929 | **.961** | .783 | .934 | **.959** |
| | $HCE_\gamma \downarrow$ | 1326 | 1116 | 1309 | 1039 | 1097 | 1802 | 1492 | **1322** | 1400 | 1412 | **1217** | 1414 | 1518 | **1057** |

## 4.2 Dichotomous Image Segmentation

Dichotomous Image Segmentation (DIS) was introduced by Qin *et al.* [24], which specifically focuses on the segmentation of objects with complex structures in high-resolution images, as shown in Fig. 1. They also built the DIS5K dataset, annotated with extremely fine details, a collection of 5,470 images with resolutions of *2K* and above.

Due to the abundance of fine-grained details in the DIS5k dataset, it poses a significant challenge for models to perform high-precision segmentation. Thus, it serves as a suitable benchmark to evaluate the proposed Pi-SAM for segmentation precision. Furthermore, for some highly challenging examples, their intricate structures are often difficult to segment perfectly in one go. Therefore, we also use this dataset to evaluate the ability of Pi-SAM to perform precise interaction.

*4.2.1 Precision Evaluation.* In this section, we evaluate the segmentation precision of the straightforward prediction (without precise interaction) of Pi-SAM. We conduct qualitative comparisons between our Pi-SAM with SAM, HQ-SAM, as well as several previous methods tailored for the DIS5K dataset, including PGNet [33],

IS-Net [24], FP-DIS[44], UDUN[22] and BirefNet[43]. The employed metrics consist of S-measure ($S_m$), F-measure ($F_\beta^x, F_\beta^\omega$), E-measure ($E_\phi^m$), Mean Absolute Error ($M$), and Relaxed HCE ($HCE_\gamma$), which are kept consistent with the previous works.

Results presented in Tab. 1 demonstrate that, firstly, our Pi-SAM significantly outperforms both SAM and HQ-SAM with a large margin. This indicates the effectiveness of our proposed additional modules and fine-tuning method. As a member of the SAM family, our Pi-SAM demonstrates significant advantages when applied to high-resolution images.

Secondly, when compared to the methods specifically designed for the DIS5K dataset, our Pi-SAM also achieves state-of-the-art performance on most metrics, with only the $HCE_\gamma$ metric being slightly lower than the previous methods. Considering that these methods underwent extensive training on the DIS5K dataset, *e.g.*, BiRefNet being trained for 580 epochs on DIS5K, they are more likely to fit the inherent distribution bias of the dataset. Therefore, the observed metric differences are acceptable.

*4.2.2 Interaction Evaluation.* We conducted this experiment to evaluate the capability of Pi-SAM to perform precise interaction.

As our objective is to evaluate the models' ability to correct erroneous predictions through interactions, while in many samples, our Pi-SAM achieves highly accurate predictions without additional interactions, leaving little room for correcting errors. Therefore, we selected a subset of the testing set of DIS5K for this experiment. Specifically, we filter out images where SAM, HQ-SAM, and our Pi-SAM have overlapping wrongly-predicted areas and then select 200 images with the poorest initial predictions as the testing samples. To automatically simulate the user clicking, we sampled points from the overlapping wrongly-predicted areas in the same way as described in Sec. 3.3. For more details about the testing images filtering and points sampling, please kindly refer to our supplementary material.

For SAM and HQ-SAM, the interaction is achieved through using the mask from the previous prediction along with new click points as the prompt for the next round of prediction. For our Pi-SAM, the interaction is achieved through the newly proposed Precise Interactor. In Tab. 2, we provide qualitative comparisons between the straightforward prediction and the prediction after interaction. Results indicate that, only our Pi-SAM achieves effective improvement in accuracy through interaction, while SAM and HQ-SAM both exhibit significant decreases in accuracy. This clearly demonstrates that our proposed Precise Interactor can effectively achieve high-precision interaction on high-resolution images, whereas this cannot be achieved through SAM's original prompt-based mechanism.

## 4.3 Salient Object Detection

In this section, we evaluate our Pi-SAM on the Salient Object Detection task, which aims to segment the most visually striking object within a scene. The employed metrics are kept consistent with Sec. 4.2.

In Tab. 3, we provide results on the two high-resolution SOD datasets: HRSOD [39] and UHRSD [34]. The comparative methods include SAM, HQ-SAM, and several baselines on these datasets, such as BiRefNet [43], PGNet [33], DHQ [27], HRSOD [39], and LDF [32]. The results demonstrate that our Pi-SAM surpasses all comparative methods across all metrics, without exception. Compared to DIS, the target objects in SOD tasks are more salient, with simpler topological structures. The outstanding performance on both DIS and SOD indicates that our Pi-SAM does not tend to show bias towards fixed target distributions and biases. On the contrary, it performs well on high-resolution images with different features.

## 4.4 Zero-Shot Evaluation

In this experiment, we report the results on the BIG dataset [5], a semantic segmentation dataset annotated on images, with resolutions ranging from 2048×1600 to 5000×3600. Since the BIG dataset is unseen by SAM, HQ-SAM and our Pi-SAM, we conduct this experiment to provide a quantitative comparison on their zero-shot capabilities. The metrics we used are the standard segmentation metric IoU and the boundary metric mBA (mean Boundary Accuracy), which are kept consistent with [5].

Results are shown in Sec. 4.4. For IoU, our Pi-SAM outperforms SAM and HQ-SAM on the testing set but slightly underperforms on the validation set. Since BIG is a semantic segmentation dataset, the

**Table 2: Results of interactively correcting wrongly predicted areas. The "Ori-x" refers to the straight-forward prediction without further interactions. For the comparisons of each pair of before and after interaction, we have bolded the improved metrics.**

| Points | Measure | Ori-SAM | SAM | Ori-HQ | HQ | Ori-Pi | Pi |
|---|---|---|---|---|---|---|---|
| 1 | $F_\beta^x \uparrow$ | .556 | .465 | .655 | .636 | .753 | **.805** |
| | $F_\beta^\omega \uparrow$ | .508 | .423 | .602 | .591 | .711 | **.773** |
| | $\mathcal{M} \downarrow$ | .246 | .315 | .156 | .179 | .106 | **.076** |
| | $S_m \uparrow$ | .572 | .493 | .673 | .66 | .758 | **.813** |
| | $E_\phi^m \uparrow$ | .659 | .584 | .734 | .725 | .832 | **.877** |
| 2 | $F_\beta^x \uparrow$ | .556 | .413 | .655 | .621 | .753 | **.809** |
| | $F_\beta^\omega \uparrow$ | .508 | .377 | .602 | .572 | .711 | **.778** |
| | $\mathcal{M} \downarrow$ | .246 | .341 | .156 | .186 | .106 | **.075** |
| | $S_m \uparrow$ | .572 | .457 | .673 | .648 | .758 | **.816** |
| | $E_\phi^m \uparrow$ | .659 | .558 | .734 | .721 | .832 | **.882** |
| 5 | $F_\beta^x \uparrow$ | .556 | .358 | .655 | .573 | .753 | **.818** |
| | $F_\beta^\omega \uparrow$ | .508 | .324 | .602 | .52 | .711 | **.792** |
| | $\mathcal{M} \downarrow$ | .246 | .36 | .156 | .196 | .106 | **.069** |
| | $S_m \uparrow$ | .572 | .42 | .673 | .614 | .758 | **.828** |
| | $E_\phi^m \uparrow$ | .659 | .535 | .734 | .694 | .832 | **.896** |

**Table 3: High Resolution Salient Object Detection results on HRSOD and UHRSD datasets. The best results among three SAM-based methods is highlighted with bold. For the comparison of all methods, top 1, 2, and 3 results are highlighted in** red **,** green **, and** blue **, respectively.**

| Methods | HRSOD | | | | UHRSD | | | |
|---|---|---|---|---|---|---|---|---|
| | $S_m \uparrow$ | $F_\beta^x \uparrow$ | $E_\phi^m \uparrow$ | $\mathcal{M} \downarrow$ | $S_m \uparrow$ | $F_\beta^x \uparrow$ | $E_\phi^m \uparrow$ | $\mathcal{M} \downarrow$ |
| LDF[32] | .904 | .904 | .919 | .032 | .888 | .913 | .891 | .047 |
| HRSOD[39] | .896 | .905 | .934 | .03 | - | - | - | - |
| DHQ[27] | .92 | .922 | .947 | .022 | .9 | .911 | .905 | .039 |
| PGNet[33] | .938 | .945 | .946 | .02 | .935 | .949 | .916 | .026 |
| BiRefNet[43] | .96 | .962 | .979 | .011 | .952 | .96 | .971 | .016 |
| SAM[14] | .932 | .955 | .963 | .022 | .88 | .913 | .921 | .054 |
| HQ-SAM[12] | .958 | .973 | .985 | .012 | .932 | .956 | .961 | .026 |
| Pi-SAM(Ours) | **.972** | **.974** | **.991** | **.006** | **.97** | **.977** | **.988** | **.007** |

**Table 4: Zero-shot capability evaluation on the BIG Test and Val Sets. The best results are highlighted with bold.**

| Model | BIG Test (100) | | BIG Val (50) | |
|---|---|---|---|---|
| | IoU | mBA | IoU | mBA |
| SAM | 0.690 | 0.673 | 0.922 | 0.720 |
| HQ-SAM | 0.825 | 0.718 | **0.930** | 0.747 |
| Pi-SAM | **0.864** | **0.766** | 0.901 | **0.788** |

annotation logic leads to several segmentation masks containing multiple discrete objects, especially in the validation set. However, our Pi-SAM misjudges the target objects in parts of this kind of data when dealing with input prompt bounding boxes, leading to some failed examples. However, for mBA, our Pi-SAM significantly surpasses the comparative methods on both sets. This result demonstrates Pi-SAM's powerful capability to predict precise boundaries and capture details on high-resolution images.

## 4.5 Ablation Study

Firstly, we conducted an experiment to evaluate the effectiveness of the proposed additional modules within our Pi-SAM. Results are shown in Tab. 5, in which the different items refer to:

(1) RGB-E: Whether the RGB embedding is incorporated into the high-resolution mask feature.
(2) HR-Conv: Whether a convolution-based output layer is employed to replace the dot-product-based output layer of SAM.
(3) Object Embedder: Whether the Object Embedder is introduced to enhance the low-resolution mask feature.

The results indicate that the introduction of each module can effectively improve the segmentation precision of Pi-SAM. Therefore, all the additional modules we proposed play crucial roles in the overall effectiveness of the system.

**Table 5: Ablation study on the incorporating of RGB Embedding and the introduction of High-Resolution Convolutional Head and Object Embedder. The best results are highlighted with bold, and our default configuration is marked in gray.**

| RGB-E | HR-Conv | Object Embedder | IoU(%)↑ | Bnd IoU(%)↑ |
|:---:|:---:|:---:|:---:|:---:|
| ✗ | ✗ | ✗ | 70.2 | 60.93 |
| ✓ | ✗ | ✗ | 79.48 | 72.48 |
| ✓ | ✓ | ✗ | 80.93 | 74.1 |
| ✓ | ✓ | ✓ | **82.52** | **76.14** |

In the second study, we evaluate the efficiency of Pi-SAM and compare it with SAM and HQ-SAM. Results in Tab. 6 indicate that, compared to SAM, our Pi-SAM introduces only a few trainable parameters while maintaining 93% of the original inference speed. Compared to HQ-SAM, our Pi-SAM introduces a similar order of trainable parameters but achieves significant precision improvements as shown in previous experiments. This effectively demonstrates the efficiency of Pi-SAM in both inference and fine-tuning.

**Table 6: Efficiency evaluation of the proposed Pi-SAM. All the three methods are the ViT-huge version.**

| Model | FPS | FLOPs(T) | Params(G) | Learnable |
|:---|:---:|:---:|:---:|:---:|
| SAM[14] | 5 | 1.49 | 1.19 | 100% |
| HQ-SAM[12] | 4.8 | 1.49 | 1.19 | 0.43% |
| Pi-SAM(Ours) | 4.65 | 1.49 | 1.19 | 0.48% |

## 5 CONCLUSION AND DISCUSSION

### 5.1 Conclusion

In this work, we explored transferring SAM into the domain of high-resolution images and proposed Pi-SAM. Compared to the original SAM and its variant, HQ-SAM, Pi-SAM demonstrates the following superiorities:

**Firstly**, Pi-SAM possesses a strong perception capability for the extremely fine details in high-resolution images, enabling it to produce high-precision segmentation masks.

**Table 7: An analysis of Pi-SAM's preference for different types of clicks during interactions. "Ori" refers to the straight-forward prediction without further interactions, "FG" refers to clicking foreground points only, "BG" refers to background only, and "ALL" refers to a combination of both.**

| Dataset | Measure | Ori | All | BG | FG |
|:---|:---:|:---:|:---:|:---:|:---:|
| | $F_\beta^x \uparrow$ | .753 | .821 | **.852** | .82 |
| | $F_\beta^\omega \uparrow$ | .711 | .8 | **.818** | .797 |
| DIS5K | $\mathcal{M} \downarrow$ | .106 | .062 | **.058** | .068 |
| | $S_m \uparrow$ | .758 | .835 | **.839** | .83 |
| | $E_\phi^m \uparrow$ | .832 | .909 | **.919** | .906 |

**Secondly**, Pi-SAM supports more precise user interactions. In addition to the native promptable ability of SAM, Pi-SAM allows users to interactively refine the segmentation predictions through simply clicking. While the original SAM fails to achieve this on high-resolution images.

**Thirdly**, Pi-SAM freezes all SAM's original parameters and introduces very few additional trainable parameters and computational costs to achieve the above performance. As a result, Pi-SAM maintains 93% of the original inference speed of SAM. This demonstrates the highly efficient fine-tuning and inference of Pi-SAM.

We believe all the experiments collectively demonstrate the powerful capability of the proposed Pi-SAM in high-resolution images, and hope that Pi-SAM can serve as a robust general segmentation tool for high-resolution images and realize its value across various downstream applications.

### 5.2 Limitation

Although the results shown in Tab. 2 demonstrate that Pi-SAM can effectively perform precise interactions with users and correct prediction errors, we found that Pi-SAM exhibits certain preferences for different types of input clicks (foreground and background), unable to equally correct foreground and background errors.

Specifically, we conduct an additional experiment to test the interaction preference of Pi-SAM. We specify three different settings where the type of clicks of each is limited to only foreground, background, or a combination of both. Results shown in Tab. 7 indicate that, Pi-SAM exhibits a preference for background clicks over foreground ones.

Upon inspection, we found that this is largely due to biases in the dataset distribution. During training, the input points for interaction are simulated based on the difference between the model's straight-forward prediction and the ground truth. The presence of numerous complex topological structures in the DIS5K dataset leads to more background prediction errors in the straight-forward prediction. Consequently, during training, the simulated interaction clicks also have a higher proportion of background. As a result, the model undergoes more training on background clicks, leading to the aforementioned model preference.

In our future work, we will continue to address the issue of imbalanced-distribution and provide users with a more interactive-friendly and high-precision segment-anything tool.

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
