# OpenReview forum: "Segment Anything with Precise Interaction"
_acmmm.org/ACMMM/2024/Conference — MM2024 Oral_

### Official Review · Reviewer_FnH6 · 2024-05-23

**Rating:** 6
**Confidence:** 3

**Summary:**

The Segment Anything Model (SAM) has achieved excellent generalization to the downstream tasks without additional finetuning. However, its performance noticeably drops in higher-resolution images for high-precision segmentation. Further, SAM fails with multiple user interactions for these higher-resolution segmentation tasks. This paper (Pi-SAM) introduces an effective approach

1) Segmenting extremely fine details in higher-resolution images.

2) Supporting more precise user interactions in higher-resolution images.

The authors have introduced two novel units to achieve these goals: a Higher-Resolution mask decoder, which includes an HR convolution head and object embedding, and precise interaction to obtain user inputs.
Instead of finetuning the SAM's modules, Pi-SAM introduces small computation overhead and optimizes them during training.

**Strengths:**

1. The paper provides a strong rationale for their proposed solution by empirically identifying the issues in SAM and existing approaches.

2. The paper presents unique ideas for decoding masks in high-definition images, using patch embeddings to capture fine details, deformable convolution to expand low-resolution masks, and precise interaction to simulate user input during training.

3. The design and implementation of precise interaction are particularly interesting and have significant potential for future research in other domains of data.

4. The paper is well-written, with detailed information and figures. The authors empirically validated their method on a diverse range of datasets. Further, the supplementary material has additional details supporting their choice of different modules in the proposed solution.

**Limitations:**

In figure2, authors may have used metrics in addition to the ground truth mask for better clarity.
I hardly find any errors in this paper.

**Suitability:**

3

---

### Official Review · Reviewer_ZVWR · 2024-05-24

**Rating:** 3
**Confidence:** 4

**Summary:**

This paper transfers the SOTA segmentation model SAM into the domain of high-resolution images with more precise user interactions.

**Strengths:**

The visualization results are promising, and the quantitative results outperform existing SOTA methods.

**Limitations:**

1. This manuscript lacks novelty. Incorporating high-level object information is a common solution to increase segmentation performance, and the ASPP structure has been well-studied. A lightweight High-Resolution Conv Head lacks specific-designed structures for solving high-resolution image segmentation problems.
2. As the authors use extra datasets to fine-tune their model, the improvement in high-resolution images may come from the extra training data.
3. I advise the authors to conduct extensive experiments on the segmentation performance on different resolutions, and discuss the available resolution range of their model.
4. The authors use object embedded to enhance the object information for better segmentation. Is this model only available for the specific objects in the training set?

**Suitability:**

3

---

### Official Review · Reviewer_WqCk · 2024-05-25

**Rating:** 4
**Confidence:** 3

**Summary:**

This manuscript addresses the challenges of high-precision segmentation in high-resolution images by proposing an enhanced version of SAM: Pi-SAM. SAM, while effective in low-resolution settings, encounters difficulties with complex structures, error correction, and boundary precision in high-resolution images. Pi-SAM overcomes these deficiencies by expanding SAM's prediction and interaction capabilities at higher resolutions, primarily through the introduction of a High-Resolution Mask Decoder and a Precise Interactor to enhance fine detail perception and error correction.

**Strengths:**

1. The figures and tables are highly informative and contribute significantly to the readability of the article. The clear and visually appealing presentation enhances comprehension and aids in conveying complex concepts effectively.
2. The authors conduct extensive experiments that thoroughly validate the effectiveness of the proposed model, providing a solid foundation for the application of high-resolution image segmentation.
3. By freezing the original SAM and introducing only a few additional parameters, the model's performance and training efficiency are significantly improved, making it a practical and resource-efficient solution.
4. The visualization results are exceptional, effectively highlighting key points and enhancing understanding. The emphasis on visual presentation helps to emphasize critical findings and contributes to the overall impact of the study.
5. The Precise Interactor module allows users to indicate wrongly predicted areas and correct errors interactively, contributing to improved segmentation accuracy and robustness.
6. The High-Resolution Mask Decoder introduces an Object Embedder and a High-Resolution Convolutional Head, effectively enhancing the mask feature's perception of target objects and correcting prediction errors. This modular approach allows for targeted feature enhancement while maintaining computational efficiency.
7. Pi-SAM efficiently models spatial correlations between pixels at different distances. This strategy enables the effective capture of detailed structures and improves the model's ability to correct prediction errors.

**Limitations:**

1. Writing error: the fourth sentence of section 2.2 is missing a punctuation mark.
2. The analysis of SAM's deficiencies in the Introduction Chapter for high-resolution images is insightful, highlighting issues with complex structures, error correction, and boundary jaggedness. However, attributing these problems solely to prediction size and feature map size may not be persuasive enough. Furthermore, even if this attribution is established, the choice to address these deficiencies by adding a High-Resolution Mask Decoder and a Precise Interactor in Pi-SAM, while shown to be efficient through experimental comparison, requires further justification. Explaining why these specific modules were chosen over other potential solutions would enhance the credibility of the argument.
3. The first section in Chapter 2 focuses on the SAM model, but it mainly discusses some basic applications of the SAM model. It is recommended to add more connections between the SAM model and this work, as well as its relevance to the motivation behind the study.
4. The introduction to the methodology could be more detailed. Specifically, it is recommended to include vital formulas in the methodology chapter to provide readers with a clearer and more accurate understanding of Pi-SAM.
5. In the interaction evaluation experiment, there are only 200 samples for testing, the explanation for selecting these samples is acceptable. However, acknowledging the potential limitations of using such a small subset and discussing the implications on the generalizability of the results would strengthen the justification.
6. The article contains many overly long sentences. Using a combination of long and short sentences would enhance the readability of the text.

**Suitability:**

2

---

### Meta-Review · Area_Chair_qpHj · 2024-06-30

**Recommendation:** Accept (Oral)
**Confidence:** 5

**Metareview:**

This paper proposes Pi-SAM, an innovative model designed to enhance the precision of high-resolution image segmentation while maintaining efficient user interaction. Pi-SAM integrates a High-Resolution Mask Decoder and a Precise Interactor module, aiming to address the limitations of the original Segment Anything Model (SAM) when dealing with fine details and high-resolution inputs.

After rebuttal and discussions, most of initial concerns are addressed. The final consensus of positive ratings lead to a acceptance for this submission.